# Toll-like Receptors and Thrombopoiesis

**DOI:** 10.3390/ijms24021010

**Published:** 2023-01-05

**Authors:** Xiaoqin Tang, Qian Xu, Shuo Yang, Xinwu Huang, Long Wang, Feihong Huang, Jiesi Luo, Xiaogang Zhou, Anguo Wu, Qibing Mei, Chunling Zhao, Jianming Wu

**Affiliations:** 1Department of Pharmacology, School of Pharmacy, Southwest Medical University, Luzhou 646000, China; 2Department of Physiology, School of Basic Medical Sciences, Southwest Medical University, Luzhou 646000, China; 3Institute of Cardiovascular Research, the Key Laboratory of Medical Electrophysiology, Ministry of Education of China, Luzhou 646000, China

**Keywords:** toll-like receptors, inflammation, thrombopoiesis, platelet, TLR2, TLR4

## Abstract

Platelets are the second most abundant blood component after red blood cells and can participate in a variety of physiological and pathological functions. Beyond its traditional role in hemostasis and thrombosis, it also plays an indispensable role in inflammatory diseases. However, thrombocytopenia is a common hematologic problem in the clinic, and it presents a proportional relationship with the fatality of many diseases. Therefore, the prevention and treatment of thrombocytopenia is of great importance. The expression of Toll-like receptors (TLRs) is one of the most relevant characteristics of thrombopoiesis and the platelet inflammatory function. We know that the TLR family is found on the surface or inside almost all cells, where they perform many immune functions. Of those, TLR2 and TLR4 are the main stress-inducing members and play an integral role in inflammatory diseases and platelet production and function. Therefore, the aim of this review is to present and discuss the relationship between platelets, inflammation and the TLR family and extend recent research on the influence of the TLR2 and TLR4 pathways and the regulation of platelet production and function. Reviewing the interaction between TLRs and platelets in inflammation may be a research direction or program for the treatment of thrombocytopenia-related and inflammatory-related diseases.

## 1. Introduction

Platelets are small (2–4 μm) and are the end products of membrane protrusions from MKs that extend into the sinusoidal vessels, where they are sheered off by blood flow [1]. Remarkably, megakaryocytes (MKs) migrate from the HSC osteoblastic niche to the vascular niche, which is spatially and temporally regulated by transcription factors, signaling and adhesion molecules, and cytokines and chemokines in the process of progressive differentiation [2]. Various aspects, such as differentiation maturity, determine the formation of platelets. Generally, megakaryocytes tailor their cytoplasm and membrane systems for platelet biogenesis. Before a megakaryocyte has the capacity to release platelets, it enlarges considerably and fills with high concentrations of ribosomes and a complete demarcation membrane system (DMS) [3,4,5]. In adults, platelet generation can be divided into two stages that entail the differentiation of HSCs into mature megakaryocytes (MKs; termed megakaryocytes) and the release of platelets from MKs (termed thrombopoiesis or platelet biogenesis) (Figure 1) [6,7]. Although human adults contain nearly one trillion platelets in circulation that are turned over every 8–10 days, it is worth noting that the reduction in platelet count and the dysfunction of platelets will lead to a variety of diseases. Thus, the relevant mechanisms driving platelet formation and platelet function are appreciated. At present, platelets have been recognized as integral players in the inflammatory response and an important mediator in the activation of innate immunity [8,9,10,11,12]. At the same time, studies have shown that platelets express a variety of receptors, some of which are involved in platelet activation, platelet–leucocyte reciprocal activation, immunopathology, and platelet-dependent antimicrobial activity, including the TLR family [13,14]. TLRs can regulate inflammation and activate the innate immune signaling pathways. Meanwhile, TLRs exist in hematopoietic stem cells and megakaryocytes, which can promote platelet generation and function (see below).

TLRs are one of the most important receptors in the activation of innate immunity and have a certain regulatory effect on the regulation of inflammation, and there is an inseparable connection between the two [15]. TLR activation triggers signaling cascades with a common downstream signaling pathway to induce the production of pro-inflammatory cytokines and type I interferons: the former triggers the synthesis of the inflammatory mediators that cause fever, pain and other inflammation, and the latter mediates antiviral responses [16,17]. In addition, studies have found that TLR’s activation of the immune response is a temporary coordinated output that can transition from releasing pro-inflammatory attack mediators to initiating the inflammatory mediators required for decomposition and tissue repair [18]. However, the abnormal activation of TLR signals can lead to autoimmune, sepsis and chronic inflammatory diseases [19,20,21]. In conclusion, TLRs are an important therapeutic or regulatory target in the inflammatory response. On one hand, the high expression of TLR on CD34+ cells suggests that the expression of TLR on HSCs is an early sensory mechanism for HSCs to directly detect infection and immediately generate immune effector cells [22]. On the other hand, the expression of TLRs is also regulated during HSC differentiation, which has also been reported previously [23,24]. It has been proven that the inflammatory factors IL-3, IL-6, IL-9 and IL-11 indirectly affect TPO-induced MK production, while IL-3 and TPO synergistically promote MK differentiation [25,26,27]. Surprisingly, as early as 1987, an increase in MK progenitor cell formation was observed in mice treated with an inflammatory agent [28]. At the same time, TLR activation produces IL-6 and tumor necrosis factor-α (TNF-α). These two cytokines have a certain effect on platelet production. A series of studies have been conducted on its mechanism [29].

These results indicate that inflammatory factors can promote platelet production and that TLR is an important receptor for regulating inflammation. Meanwhile, platelets also express TLRs, and activating this receptor can regulate the function of platelets. In addition, TLR2 and TLR4 are the main receptors associated with platelet production/function and inflammation in the TLR family [30]. Moreover, some drugs acting on the TLR2 and TLR4 signaling pathways have been developed to treat diseases in order to improve the probability of cure and quality of life [31]. Therefore, we discuss the relationship between platelets and inflammation, as well as TLR regulatory pathways, and summarize the potential role of TLR on platelets in inflammation and the development of drugs in related pathways.

## 2. Inflammation and Platelets

Inflammation, as the second line of defense in innate immunity, is the body’s defense mechanism against pathogens, allergens/irritants, damaged cells, or any foreign invaders, manifested as redness, swelling, heat, pain, or dysfunction [32,33,34]. It is also an important part of innate immunity. At present, studies have shown that the body activates different pathways in response to the relevant stimuli and generally releases a large number of inflammatory factors, primarily γ-interferon, interleukin (IL), nitric oxide (NO) and TNF-α. It is beneficial to the body to remove harmful signals and initiate the response of the tissue healing process. Generally, it can be divided into two types, chronic and acute, depending on the characteristics of the response and stimulation. Among them, acute inflammation is primarily a form of regulation to restore homeostasis after damage to the body tissue, the symptoms of which are fever, redness, swelling and pain. The body response mainly involves mast cells releasing histamine and prostaglandins to dilate their blood vessels, and granulocytes migrate quickly to neutralize and remove harmful substances [35]. However, if acute inflammation fails to restore homeostasis, acute inflammation will gradually evolve into chronic inflammation and even cause more diseases. In chronic inflammation, macrophages and T lymphocytes mainly produce cytokines and enzymes that lead to tissue destruction, manifested as tissue fibrosis. As vascular first responders, platelets are key elements in the early phases of the inflammatory response (Figure 2). Platelets in the human body are not a uniform population, and platelet content varies with their size. Smaller platelets carry more inflammatory transcripts than larger platelets in a healthy population [36]. During the period of infection, antiviral protein in the blood platelet increases because the megakaryocyte cells start the reaction in the process of infection as a response to pathogens, directly activating platelets and releasing extracellular vesicles. In addition, platelets can directly sense pathogens and endogenous danger signals and promote the formation of antimicrobial and procoagulant neutrophil extracellular traps (NETs) through the expression of TLRs [37]. Remarkably, at the same time, platelets help leukocytes infiltrate tissues and exert their immune functions; they limit host tissue collateral damage by repairing vascular breaches caused by leukocyte trafficking and activation [37]. This suggests that platelets are an important mediator between innate immunity and inflammation [38]. In addition, the pathogenesis of inflammatory disease models has been demonstrated to rely on platelet-neutrophil complex formation [39]. First, they activate macrophages and monocytes and are critical in the recruitment and activation of neutrophils and they collaborate with neutrophils to initiate intravascular thrombosis, contributing to pathogen recognition, trapping, and disposal, thereby protecting the integrity of the host [39]. However, intravascular thrombosis caused by the interaction of platelets and white blood cells is called immune thrombosis, and this thrombotic reaction lasts too long in the body and may lead to disseminated intravascular coagulation (DIC), followed by multiple organ failure and even death. Second, platelets can migrate, aggregate, and adhere to bacteria in the body, contributing to the recruitment and stimulation of neutrophils and thus playing a direct antibacterial role [40,41]. This makes platelets attractive therapeutic targets for diseases ranging from chronic to acute inflammation or autoimmune diseases to bacterial, viral, and parasitic infections [42,43]. However, it is worth noting that, whilst platelets are the link between inflammation and host defense, the overreaction of platelets can also lead to disease. This is sufficient to show the multiplicity of platelets in inflammation.

In addition to some host-protecting reactions of platelets during inflammation, there are also studies showing that inflammation also plays a certain role in regulating platelet production. Thrombopoietin (TPO) is the most important regulator of megakaryocyte differentiation and platelet production [44]. In addition, studies have found that many inflammatory factors, such as IL-3, IL-6, IL-9, IL-11, IL-1α and IL-1β, can promote the role of TPO in promoting platelet production, while IL-3 and TPO can synergistically promote MK differentiation [25,26,27,45,46,47]. These results suggest that in the inflammatory state, inflammatory factors may be involved in mediating megakaryocyte differentiation and promoting platelet generation by regulating TPO synthesis [48,49]. In addition, IL-6 contributes to driving thrombocytosis in systemic inflammation during the acute inflammatory response [47,50]. For megakaryocyte progenitors, studies have also shown that inflammatory signals activate posttranscriptional protein synthesis, maturation and cell cycle induction in quiescence but initiate stem-like megakaryocyte progenitors (sl-MKPs) to improve megakaryocyte formation to replenish the platelet pool during acute inflammation, while sl-MKPs remain in the activated but quiescent state in the homeostatic state [51]. In addition, as early as 1987, after treating mice with inflammatory preparations, it was observed that the formation of MK progenitor cells increased [28]. The further link between inflammation and megakaryopoiesis can be the ROS (reactive oxygen species). When the body is endangered, innate immunity is activated, in which macrophages and neutrophils are activated and release ROS. At this time, HSCs will develop a bias toward the megakaryocyte lineage [52]. In addition, studies have also shown that in acute inflammation, IL-1α leads to the rapid, TPO-independent production of platelets, primarily through IL-1α signaling, decreased plasma membrane stability, dysregulated tubulin expression and proplatelet formation, which eventually trigger megakaryocytes to rupture and release large numbers of platelets in a short period of time [53]. According to research, megakaryocytes, the major source of platelets, express TLR, tumor necrosis factor receptors (TNFR1 and 2), IL-1β and IL-6 receptors, and the activation of the above receptors can lead to inflammatory factors, chemokine release and the activation of the NF-κB channels, which link inflammation with platelet formation [54,55,56,57]. TLRs are a crucial family in the inflammatory response and innate immunity. Once activated, the downstream pathway releases inflammatory factors and regulates the process of platelet generation; meanwhile, TLRs are also expressed on platelet surfaces. Therefore, it is very important to understand the relationship between TLRs and platelets and inflammation, which has certain prospects for the study of thrombocytopenia or the treatment of inflammatory diseases.

## 3. Inflammation and TLRs

It is well-known that the inflammatory response triggered by infection or tissue injury involves the coordinated delivery of blood components (plasma and leukocytes) to the site of infection or injury. This response has been best characterized for microbial infections (particularly bacterial infections), in which it is triggered by receptors of the innate immune system, such as TLRs and NOD (nucleotide-binding oligomerization-domain protein)-like receptors (NLRs) [58]. The mammalian Toll-like receptor family, named for its similarity to the toll gene first described in Drosophila, is highly conserved in a variety of organisms [59]. TLR is one of the major pattern recognition receptors (PRRS) expressed by immune and non-immune cells (including neurons) and plays a key role in generating cytokine storms. It is primarily a class of proteins that can recognize conserved molecular signals on pathogen-associated molecular patterns (PAMPs) and danger-associated molecular patterns (DAMPs) [60]. At the same time, it is a pattern recognition receptor of the innate immune system, and its activation is also the host’s first defense against the invasion of microbial pathogens (Figure 3). Notably, exogenous danger signals, commonly called PAMPs, are highly conserved motifs in microbial pathogens, whereas DAMPs are derived from damaged and injured cell proteins, cytokines, chemokines and other molecules. After recognizing PAMPs and DAMPs, TLRs activate the downstream signaling pathways that release a variety of pro-inflammatory mediators (cytokines, chemokines, interferons, ROS and nitrogen species (RNS)), which cause acute inflammation to the control pathogens and repair damage. However, sometimes an overreaction due to the genetic makeup of the host and/or the persistence of the pathogen due to its escape mechanism can lead to severe systemic inflammatory conditions in response to cytokine storms and the generation of organ dysfunction [61,62]. The activation of the TLR-induced inflammatory response is closely related to the induction of several negative feedback mechanisms that function to end the inflammatory response and maintain immune homeostasis [63]. These results are sufficient to indicate that the activation of TLRs is closely related to the production of inflammation. Current studies have shown that PAMPs recognized by TLRs include lipids, lipoproteins, proteins and nucleic acids, which originate from a wide range of microorganisms, such as bacteria, viruses, parasites and fungi [60]. While there are 13 TLRs in humans and mice, among these, ten are known functional human TLRs (TLR1–10), and twelve are functional mouse TLRs (TLR1–13, except TLR10) [61,62]. TLRs can be divided into two subgroups, according to their cellular localization: one group is predominantly expressed on the cell surface and primarily recognizes microbial membrane components such as lipids, lipoproteins and proteins (TLR1–2, TLR4–6, TLR10); the other group is only expressed in intracellular vesicles, such as endoplasmic reticulum (ER) lysosomes, and primarily recognizes microbial nucleic acids (TLR3, TLR7–9) [19,61]. In addition, some studies have clearly indicated that TLR1, TLR2, TLR4 and TLR6 can be expressed on megakaryocytes, while platelets express TLR1, TLR2, TLR4, TLR6, TLR7, TLR8 and TLR9 [34,64,65,66,67,68,69,70]. Human CD34+ progenitor cells express TLR4, TLR7, TLR8, and TLR9, and, in vitro, the stimulation of TLR7 and TLR8 induces the myeloid differentiation of these progenitors [71]. Each TLR is specific and can recognize specific molecular features, and the key ligands of TLRs are shown in Table 1.

TLRs are type 1 transmembrane proteins defined by an extracellular leucine-rich repeat (LRR) domain for ligand recognition, a transmembrane domain (TMD) and a cytoplasmic Toll/interleukin 1 receptor (TIR) homology domain [72]. The extracellular LLR domain is primarily responsible for ligand binding, while the TIR domain binds to aptamer molecules [46,73,74]. Its main ligand adaptors can be divided into two types: typical and regulatory, as shown in Table 1. Researchers have found that the TLR activation pathway plays an important role in balancing pro-inflammatory and anti-inflammatory cytokines [21,75]. The signaling pathways activated by TLRs can be roughly divided into two categories: one is the MyD88-dependent pathway, which drives the induction of inflammatory cytokines; the other is the TRIF-dependent pathway, which is responsible for the induction of type I interferons and inflammatory cytokines [6,76,77]. MyD88 is the first TIR-containing adaptor protein found to be involved in all known TLR signaling pathways, with the exception of TLR3, and a cytoplasmic adaptor and mice lacking functional MyD88 are noted for their unresponsiveness to bacterial endotoxin [60]. It is predominantly composed of three functional domains: the death domain, intermediate domain and C-terminal TIR domain. After the engagement of TLRs by their cognate PAMPs, MyD88 interacts with TLRs through its TIR domain (a TIR–TIR interaction) and recruits downstream interleukin-1 receptor-associated kinase 4 (IRAK4) through a homotypic death domain protein–protein interaction [78]. The subsequent recruitment of IRAK1, IRAK2 and TNF receptor-associated factor 6 (TRAF6) by IRAK4 results in the formation of a ‘myddosome’ complex [79,80,81]. IRAK4 is initially activated and has an essential role in the activation of NF-κB and MAPK, downstream of MyD88. IRAK1 and IRAK2 are activated sequentially, and the activation of both kinases is required for the robust activation of NF-κB and MAPK. IRAK activation results in an interaction with TRAF6, an E3 ligase that catalyzes the synthesis of polyubiquitin linked to Lys63 (K63) on target proteins, including TRAF6 itself and IRAK1, in conjunction with the dimeric E2 ubiquitin-conjugating enzymes Ubc13 and Uev1A. The K63-linked polyubiquitin chains then bind to the novel zinc finger-type ubiquitin-binding domain of TAB2 and TAB3, the regulatory components of the kinase TAK1 complex, to activate TAK1, which activates different signaling pathways, leading to nuclear translocation of the transcription factor NF-κB and transcription of NF-κB and AP-1-dependent pro-inflammatory cytokines such as tumor necrosis factor (TNF) and IL-6. Significantly, the myddosome is a precise, stoichiometric assembly of MyD88 with its downstream IRAK that is nucleated in response to an activated receptor complex for signal transduction, and in the case of TLR4, the signal intensity relies on the speed of the myddosome assembly, as well as the myddosome number and size, as revealed by single-molecule imaging [82]. However, TRIF-dependent signaling, which is an independent branch of TLR signaling, is maintained only by TLR3 and TLR4 [83,84]. In this pathway, TRIF interacts with TRAF3 and TRAF6, but TRAM is thought to interact with TRIF to promote the activation of the TRAF3-dependent kinase TBK1, which then drives IFN induction and interferon-stimulated gene expression. In summary, after different ligands bind to TLRs, the cytoplasmic TIR domain of TLRs recruits the signaling adapters MyD88, TIRAP, TRAM and/or TRIF, which activate different downstream pathways according to different adapters, such as Type I interferons, p38 MAPK and JNK MAPK pathways (Figure 4).

## 4. TLRs and Platelets

The inflammatory and immune functions performed by platelets can be explained by the fact that, in lower vertebrates such as fish and birds, platelets and white blood cells share a common ancestor cell, the thrombin cell, which has hemostatic and immune functions. As immune cells, one of the most relevant characteristics of platelets is the expression of TLRs. TLRs are key receptors that transmit danger signals to the innate immune system and mediate inflammatory events, ultimately recruiting and activating cells of the adaptive immune system to respond to the invading pathogens. In short, TLRs are both innate immune receptors and important receptor components that mediate the inflammatory responses. Platelets express a variety of TLRs, which can elicit a plethora of responses, including activation, homotypic and heterotypic aggregation, the release of inflammatory mediators and the modulation of the leukocyte inflammatory responses. Platelets are also key secretors of TLR-triggering DAMPs and, in this capacity, serve as initiators of innate immunity. Moreover, some TLRs are also expressed in megakaryocytes, and their activation not only regulates platelet biogenesis but also regulates pro-inflammatory and anti-viral responses. TLRs therefore play a central role in linking inflammation and platelet formation/function.

### 4.1. TLR2 and TLR4 Signaling Pathways

TLR2 and TLR4 represent two important types of these receptors, mediating intracellular activation through the MyD88-dependent pathway. TLR4 is the first TLR that exists and plays a role in platelets, and it is also the most abundant TLR expressed in platelets [68,85]. In addition, TLR2 and TLR4 expressed on platelets are far superior to other TLRS and are also the most studied receptors at present. Meanwhile, it is surprising that TLR2 and TLR4 are not only important components of bacterial pattern recognition but also important receptors regulating megakaryocyte differentiation, as well as platelet production and function. TLR2 is involved in the recognition of a wide range of PAMPs from bacteria, fungi, parasites and viruses [6]. TLR2 usually forms heterodimers with TLR1 or TLR6, but the two differ in their recognition of signaling molecules. TLR2-TLR1 heterodimers can recognize triacylated lipopeptides in gram-negative bacteria and mycoplasma, while TLR2-TLR6 heterodimers are dimers that recognize acetylated lipopeptides, mainly because TLR6 does not have a hydrophobic pocket to bind [86,87]. However, the TLR2-related signaling pathway is roughly the same as that of TLR4, except that there is no TRIF-dependent signaling pathway, namely, IFN induction and IFN-stimulated gene expression.

Some of the most important TLR4 ligands include LPS, viral motifs and DAMPs such as soluble hyaluronan, beta-defensin 2, high-mobility group box 1 (HMGB1) protein and histones [88]. TLR4 is primarily a receptor for the production of bacterial lipopolysaccharide (LPS), a component of the outer membrane of gram-negative bacteria that can cause septic shock [60]. TLR4 can induce distinct signaling pathways from two different organelles [89]. The first signaling pathway is activated from the plasma membrane, mainly by a pair of sorting and signaling adaptor proteins called TIRAP and MyD88, respectively, and these linker interactions can be recruited to TRAF6, resulting in three main pathways [90,91,92,93]: ① TRAF6 is modified by K63-linked autoubiquitination, which enables IκB kinase (IKK) to pass through the ubiquitin binding domain of the IKKγ (also known as NEMO) subunit to activate the NF-κB pathway; ② the ubiquitin-binding domain of TAB2 recognizes ubiquitinated TRAF6, leading to the activation of the associated TAK1 kinase, which then phosphorylates the IKKβ subunit, which activates the NF-κB and JNK pathways; ③ TRAF6 modifies cIAP1 or cIAP2 with K63-linked polyubiquitin so that cIAP is activated to modify TRAF3 with K48-linked polyubiquitin, leading to its proteasomal degradation and allowing the TRAF6-TAK1 complex to activate the p38 MAPK pathway and promote the production of inflammatory cytokines. Notably, TLR4 forms a complex with myeloid differentiation protein 2 (MD2) on the cell surface to recognize foreign substances and activate the downstream channels [94]. For example, when TLR4 recognizes LPS, LPS has six lipid chains, five of which bind to the hydrophobic pocket of MD2, and the remaining lipid chains bind to TLR4 [95,96]. Studies have shown that CD14, a glycosylphosphatidylinositol-linked leucine-rich repeat protein, is important for TLR4-dependent MyD88-dependent signaling. It primarily binds to LBP (LPS-binding protein), LBP binds LPS and then binds to CD14, and CD14 can transport the LPSLBP complex to the TLR4-MD2 complex, thereby activating the downstream pathway [32]. The second is the internalization of TLR4 into the endosome network, which is mainly triggered by the adaptors TRAM and TRIF, promoting the activation of the TRAF3-dependent kinases TBK1 and interferon regulatory factor-3 (IRF3) to regulate the expression of type I interferon (IFN) [83,88,97]. Notably, BCAP can link TLR4 signaling to PI3K activation. PI3K–AKT activation has regulatory effects on the outcome of TLR signaling, including limiting pro-inflammatory cytokine secretion and promoting anti-inflammatory cytokine production, which is unique to TLR4 [98].

### 4.2. TLR2 and TLR4 Promote Platelet Formation

The TLR family plays an important role in the innate and adaptive immune responses and can sense exogenous pathogens and endogenous inflammatory stimuli [99]. Meanwhile, the relationship between inflammation and TLRs, which has a significant impact on the regulation of the balance of inflammation, has been confirmed [17,100]. In addition, a wide range of endogenous TLR activators, such as heat shock protein and HMGB1, have been observed in the synovium of patients with rheumatoid arthritis (RA) but not in the non-inflammatory synovium of patients with normal joints or osteoarthritis (OA) [101,102,103]. Different ligands that activate the corresponding TLR pathway produce inflammatory factors. For example, when TLR2 is activated by its related ligands, the NF-ĸB signaling pathway is activated, which further leads to the increased expression and release of inflammatory factors such as interleukin and TNF-a [104,105]. Notably, IFN-1 and TNF-α can rapidly activate the post-transcriptional megakaryogenic program in a subset of hematopoietic stem-like cells expressing high levels of CD41 [106,107]. A similar phenomenon occurs during chronic IL-1 stimulation, resulting in increased platelet production during this period of inflammation [51]. Therefore, researchers have launched a series of studies on TLR family representatives, namely TLR2 and TLR4, which are currently the most studied, and megakaryocyte differentiation.

First, in in vitro studies, some studies have clearly shown that Pam3CSK4 activates TLR2 to induce the following cascade reactions: (1) NF-κB, ERK-MAPK and PI3K/AKT pathway activation; (2) increased levels of transcription factors related to megakaryocyte maturation increased, such as mTOR, GATA-1 and NF-E2; and (3) enhanced the gene expression of CD61, CD41, MCP-1, COX2, NF-κB1 and TLR2, and increased the levels of inflammation-related proteins [108]. Notably, CD61 and CD41 are both markers of megakaryocyte maturation. This finding directly indicates that TLR activation can promote megakaryocyte maturation and platelet formation through the downstream pathways. When TLR4 is activated by its natural agonist LPS, it leads to the activation of JAK-2 and STAT-5 and induces macrophages to secrete the inflammatory cytokine IL-6, which is also a positive regulator of megakaryocyte formation [109,110]. In addition, TLR2 activation can also stimulate Meg01, which can promote the activation of the TLR downstream pathways and subsequent megakaryocyte maturation [64]. When Dami cells were treated with heat-killed lactic acid bacteria HKL (1 g/mL), another TLR2 ligand, for 24 h, their expression levels were detected by qPCR, which showed that the expression of TLR in the stimulated group increased 5-fold, and the expression of megakaryocyte maturation markers CD41 and CD61 also increased [54]. In addition, the level of ROS was also increased by flow cytometry, and these effects were mediated through the activation-catenin component, indicating that the Wnt and TLR pathways are connected. The crosstalk leads to the maturation of megakaryocytes and promotes the release of cytokines, of which IL-6 is the most important. However, IL-6 was able to enhance the effect of TPO on promegakaryocyte differentiation and myelofibrosis progression [111,112]. We observed an increased expression of the cellular fibronectin EDA isoform (EDA-FN) and the EDA fibronectin–TLR4 axis induced megakaryopoiesis through profibrotic IL-6 release. In addition, TLR-2-induced megakaryocyte-like particles (MKEVs) recapitulate the TLR-2 signaling pathway that activates megakaryocytes and may increase production to induce the differentiation and maturation of megakaryocytes [113]. The activation of TLR2 promoted Meg-01 cell differentiation and increased CD41 and GPIb (glycoprotein receptor Ib) mRNA and protein levels [64]. In a study of zymosan and/or LPS-treated Dami cells, the expression of megakaryocyte markers (CD41 and CD61) and the level of NF-ĸB protein were also increased in costimulatory treatment [114]. In addition, the activation of TLR2 and TLR4 in CD34+ cells and megakaryocytes in the presence of TPO may warrant platelet provision during infection episodes by an autocrine IL-6 loop triggered by the PI3K/NF-κB axes [115].

The researchers also carried out a series of in vivo studies. First, TLR2- or TLR4-deficient mice have lower platelet counts than wild-type mice, indicating that these TLRs play a role in platelet production, so they can be predicted to promote thrombopoiesis [64]. In addition, the hypothesis that TLR4 activation promotes megakaryocyte differentiation and platelet production was reinforced in a study in which mice stimulated with TLR4 ligands had higher platelet counts at nonlethal doses [116]. Meanwhile, several studies have found that during embryogenesis in mice and zebrafish, pro-inflammatory signaling through TLRs promotes the emergence of HSPCs in hematopoietic endothelial cells [117]. The stimulation of TLR4 in hematopoietic endothelial cells generates inflammatory signals through Notch activation to promote HSPC development, while the loss of TLR4 signaling results in a marked reduction in the appearance of HSPCs [118]. HSPCs are the main source of megakaryocytes, which further indicates that there is a certain connection between inflammation and TLRs and megakaryocytes. Furthermore, an increase in spleen size and extra-medullary hematopoiesis could be counteracted by treatment with an inhibitor of TLR4 (TAK-242), and a slight reduction in reactive peripheral thrombocytosis, leukocytosis, and anemia was also observed after TAK-242 treatment [112]. Various studies have suggested that future medical drug development research can utilize the TLR signaling pathway to treat symptoms of thrombocytopenia and inflammatory diseases but, at the same time, it is necessary to balance the relationship between TLR pathway activation and inflammation (Figure 5).

### 4.3. TLR2 and TLR4 Increase Platelet Function

It is well known that platelets are enucleated blood cells involved in a variety of physiological and pathological functions, and their main role is to mediate the rapid activation-dependent response of hemostasis and wound repair. In addition to these classical functions, platelets have emerged as important players in the innate immune system through many functions that last hours after pro-hemostatic adhesion and aggregation [12,119]. It also expresses TLRs, mainly TLR1/2/4/6, on the surface of cells and TLR3/7/9 in the endosome [66,69,120,121] The detection of PAMPs is therefore an effective host defense function of platelets, ensuring a rapid response to infection [122]. Furthermore, although platelets contain intracellular signaling proteins required for TLR signaling and do not have all of the necessary membrane CD14 for nucleated cells, high levels of soluble CD14 in plasma overcome this problem [85,123,124,125]. Thus, the activation of TLR2 and TLR4 on other cells promotes platelet production, while the activation of TLR2 and TLR4 on platelets can regulate their functions, such as anti-infection, adhesion, and aggregation (Figure 6).

Just as the activation of TLR on other cells releases different cytokines, so does the activation of TLR on platelets. In addition, studies have shown that TLR4 activation can increase platelet-neutrophil aggregates, neutrophil extracellular trap (NET) formation (NETosis), and bacterial capture in sepsis [126]. At the same time, when platelets were cocultured with neutrophils and TLR4 agonists, neutrophil CD62 L expression, phagocytosis and IL-8 secretion were increased, but CD62 L shedding and elastase secretion were decreased [127]. TLR2 on platelets can also recognize bacterial lipopeptides together with TLR1 and TLR6 to stimulate and induce the formation of platelet-neutrophil aggregates and bacterial phagocytosis [65,74,128]. All of the above results indicate that the activation of TLR2 and TLR4 can promote the anti-infective effect of platelets. Studies have shown that Pam3CSK4, the ligand of TLR2, also controls the increase in the Ca2+ concentration, ATP release and TXA2 synthesis in platelet cells [129]. In addition, studies have shown that Pam3CSK4 directly induces platelet aggregation through the classical activation of TLR2, while LPS-activated TLR4 indirectly induces platelet aggregation by enhancing platelet adhesion in a GPIb-dependent manner, which may increase thrombus formation [130,131]. Notably, Pam3CSK4 induced platelet aggregation to a lower degree than in response to thrombin; however, stimulation with this agonist resulted in significant platelet-monocyte aggregation that was not seen with thrombin [132]. It is worth noting that the above TLR2 agonistic responses are all directed against the binding of TLR2/1, while in most studies, stimulation with the prototype TLR2/6 agonist cannot induce platelet aggregation or activated aggregation [65,129,133]. However, at supraphysiological doses, another canonical TLR2/6 agonist, fibroblast-stimulating lipopeptide-1 (FSL-1), induces platelet activation and heterotypic aggregation in whole blood but not in platelet-rich plasma (PRP) [134]. Meanwhile, in a study of TLR activation, platelet activation was only responsive to supraphysiological doses of LPS (100 μg/mL), but not to lower doses (0.1 to 50 μg/mL) [135].

In short, the TLR2 and TLR4 signaling pathways not only play a role in the platelet formation process but also have a certain correlation with platelet aggregation, activation and interaction with leukocytes. Therefore, TLR can be considered a target for the treatment of platelet function-related diseases but, at present, there are few studies on this kind of correlation, which require further studies.

### 4.4. Balanced TLR2 and TLR4 Pathways Are Critical

Although a large number of studies have shown that the activation of the TLR2 and TLR4 signaling pathways can promote the formation and function of platelets, many studies have shown that the over-activation of this pathway leads to the occurrence of some diseases, deterioration and even life-threatening conditions [136]. In particular, undue TLR stimulation may disrupt the fine balance between pro- and anti-inflammatory responses and may harm the host through the development of autoimmune and inflammatory diseases, such as rheumatoid arthritis and systemic lupus erythematosus. This finding reminds us that inflammatory diseases can be treated with TLR pathways. For example, reducing the activation of the TLR2 and TLR4 signaling pathways can reduce inflammation and promote neurological recovery after focal cerebral ischemia, and vice versa [137]. Meanwhile, the excessive activation of the TLR signaling pathway can lead to the release of a large number of inflammatory factors, and the enhanced release of cytokines after the inflammatory response can increase the risk of venous thrombosis [138]. Further studies have shown that activation of the TLR4 pathway leads to late activation of NF-kB and toxicity of TNF-α, thereby increasing the risk of thrombosis [139,140]. In conclusion, the downstream pathways activated by TLR2 and TLR4 include positive feedback loops, and TLR4 expression levels are associated with more severe disease states in inflammatory responses, such as neuroinflammation, cardiovascular disease, and cancer [141,142,143,144,145]. Therefore, TLR signaling pathways must be tightly regulated (Figure 7).

TLRs induce the phenotypic and secretory activation that interacts with endothelial and immune cells to contribute to the inflammatory response and subsequent repair process, while also promoting an increase in a component when activated under certain conditions, such as platelets. However, whether their post-activation effects are beneficial or detrimental depends not only on their micro-environment, including the cytokine milieu and the activation state of their cellular partners but also on the nature of the activating factors; thus, platelets may contribute to the dysregulation of the inflammatory response and the immune escape of pathogens, whether microbes or tumors, but they may also produce damage that repairs the body. Therefore, when using the TLR signaling pathway to treat thrombocytopenia or control the inflammatory response, we must pay attention to the balance of the regulation of the TLR signaling pathway to maximize the advantages and minimize the disadvantages.

### 4.5. Other TLRs and Platelets

It is now well known that platelets express multiple TLRs. As mentioned above, TLR1/6 can combine with TLR2 to activate the downstream pathways, and it has been reported that it promotes platelet production and affects platelet function. Poly(I:C), a synthetic dsRNA replication product and TLR3 ligand, activates TLR3 to reduce platelet preformation, as well as release and affect its function [146]. Some studies have reported that the activation of TLR3 may lead to the production of IFN-β, which in turn reduces the production of platelets [147]. It has also been shown that although Poly(I:C) and poly(A:U) do not directly trigger platelet aggregation, they can potentiate the binding of fibrinogen, as well as the aggregation and the release of ATP mediated by classical agonists such as thrombin, ADP, collagen and arachidonic acid [120]. However, how the expression of TLR3 affects the production and function of platelets still requires further study. The role of TLR7 in platelet production has not been extensively studied, but in patients with primary Sjogren’s syndrome-related thrombocytopenia, TLR7 and its downstream signaling molecules are strongly expressed [69]. However, the specific mechanism is currently not clear. In addition, in a study of TLR7 in human and mouse platelets, it was found that only platelet TLR7 activation initiated the formation of heterogeneous aggregates between platelets and granulocytes, thereby increasing the adhesion between platelets and neutrophils [69]. This suggests that platelets are the blood cells that initiate communication with neutrophils only when TLR7 is activated. However, it is worth noting that the activation of TLR7 does not promote the platelet aggregation and platelet activation caused by thrombin. These results all indirectly demonstrate that direct thrombotic events appear to be independent of platelet TLR7. While TLR9 is expressed in all stages of thrombopoiesis, it not only recognizes viruses but also senses endogenous ligands produced under oxidative stress-related pathophysiological conditions, such as carboxyalkylpyrrole protein adducts (CAPs), which are proteins modified by lipid peroxidation [70]. This protein promotes platelet activation and aggregation in vitro and accelerates thrombosis in vivo in a TLR9/myd88-dependent manner. In addition, TLR5 expression on platelets has recently been found to be increased in thrombin-activated and bacterial sepsis patients [148]. Meanwhile, TLR8 mRNA in patients with rosacea bacterial chitin stimulation showed an increase in platelets, whereas other TLR mRNAs were not regulated, suggesting a role in platelet TLR8 responses to Candida [149].

In conclusion, the different signal transduction pathways and effector molecules triggered by the activation of different TLRs seem to determine the positive or negative fate of platelet production, respectively. At the same time, the activation of TLRs under the relevant conditions also affects platelet function, thereby regulating human homeostasis and disorder or contributing to the progression of human disease. Therefore, it is meaningful to clearly study the relationship between TLRs and platelets for the medical treatment of diseases.

## 5. Conclusions and Future Perspectives

As research into the TLR family continues, human and animal genetic studies have shown that the dysregulation of innate immune TLR signaling contributes to the development and progression of various diseases, including sepsis, autoimmune disease, and neuropathic pain. Therefore, some studies have found that some drugs also rely on this TLR-related signaling pathway to achieve the best efficacy. For example, a new type of ginseng-derived nanoparticle (GDNP) can change the polarization of M2 macrophages in vitro and in vivo, which largely relies on TLR4 and MyD88 signaling to enhance antitumor responses [142]. In addition, some biotech and pharmaceutical companies have been actively involved in the research of TLR drugs to adjust the state of the body; that is, either agonists to correct insufficient immune responses or antagonists to suppress excessive activation (see Table 2 and Table 3). Overall, the TLR signaling pathway is a double-edged sword, playing a dual role as a physiological and pathological mediator. Therefore, when developing related drugs, we must grasp the signal strength of this pathway. The current preclinical and clinical breakthroughs in this classification of drugs may improve the availability of TLR immunomodulatory drugs to address important unmet medical needs. For example, the TLR4-specific antagonists can reduce the overproduction of inflammatory mediators to inhibit neuritis [150]. In general, TLR agonists are used as immunotherapy or vaccine adjuvants for the treatment of cancer, allergies and infectious diseases, while TLR antagonists can be potential treatments for chronic inflammatory and autoimmune diseases by tightly regulating the overactive immune response strategies. Currently, an increasing number of TLR-related drugs are being developed, and they do have potentially relevant clinical indications based on the preclinical data. For example, TLR4 inhibition can be used to treat arthritis and reduce central and peripheral neuralgia, post-traumatic stress disorder (PTSD) and major depressive disorder (MDD) [151,152,153,154,155]. Although there is now a growing list of potential clinical indications regarding the broad role that TLR4 plays in immune- and inflammation-mediated pathological conditions, the clinical data on TLR4 antagonism are limited. Only E5564 and TAK-242 have been studied in large randomized trials for the treatment of sepsis, with results demonstrating acceptable safety and tolerability [156,157,158]. Meanwhile, E5564 also showed limited efficacy in a phase II trial in patients undergoing cardiopulmonary bypass, but was well tolerated [159]. In addition, bacille Calmette-Guérin (BCG, TLR2/TLR4 agonist) and monophosphoryl lipid A (MPL, TLR4 agonist) have been approved by the U.S. Food and Drug Administration (FDA) for the clinical treatment of cancer patients [160]. Remarkably, TLR-related agonists have only been confirmed in pre-clinical studies, and there are few post-clinical studies, especially in the field of thrombocytopenia, which is worth considering. However, it is worth noting that there are few studies on the development of TLR-related agonists to treat diseases, their inhibitors have only been confirmed in pre-clinical studies, and there are few post-clinical studies. In addition, there are few pre-clinical/post-clinical trials to obtain reliable data in the field of thrombocytopenia or platelet function disorders, which is worth taking into consideration.

Finally, according to inflammation, platelets are the first key players in the vascular response and can be activated to connect different cells to carry out a host protective response. In addition, inflammation can also promote the formation of platelets in hematopoietic cells. The activation of the TLR family not only regulates the inflammatory response but is also one of the important receptor families in the innate immune host defense. Meanwhile, studies have proven that platelets are important mediators in the connection between inflammation and innate immunity. Therefore, in this paper, we found that the TLR family plays a variety of different roles in platelets. On the one hand, expressed in the lineage of platelets, it can release inflammatory factors or combine with other pathways to promote platelet formation and release; on the other hand, expressed on the surface of platelets, it can be activated to release different cytokines and can aggregate/recruit/activate other cells for the host defense response. However, it is worth noting that the excessive activation of TLRs can lead to chronic inflammation and autoimmune diseases, while TLR defects can lead to cancer and allergies. Therefore, it is very important to balance the signal strength and make good use of this pathway for drug development. However, there are many challenges in developing drugs and balancing TLR signaling. First, TLRs are widely distributed in body tissues and located on the cell membrane/surface and nucleus; therefore, ensuring the localization of their activation to achieve the desired efficacy is a primary challenge. Second, how to control the activation of signaling pathways to make it more precise is also an important factor for researchers to consider in drug development. Finally, there are many signaling pathways in the human body, not only the TLR signaling pathway, which can lead to crosstalk between signaling pathways. Therefore, in the activation or inhibition of the TLR signaling pathway, it is critical to ensure that other signaling pathways are not affected or the impact is too small to damage the human body. At present, research on TLRs is still limited, which calls for a deeper exploration of the TLR signaling pathway, its cross-pathways and more preclinical studies related to platelets. Only in this way can we overcome the various challenges and open up new approaches for the treatment of platelet-related clinical diseases.

## Figures and Tables

**Figure 1 ijms-24-01010-f001:**
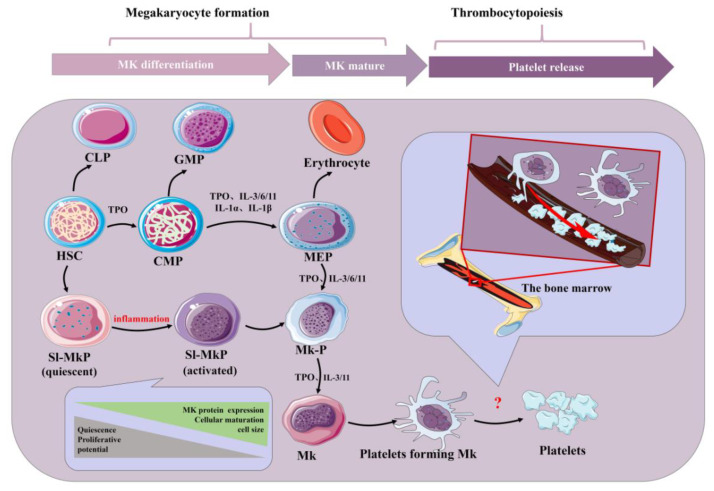
Schematic of platelet production. Inflammatory factors play an important role in the formation of platelets by bone marrow hematopoietic stem cells under physiological and inflammatory conditions. Acute inflammation can drive efficient cell cycle activation and maturation of SL-MKPs to rapidly replenish platelet pools. HSC: hematopoietic stem cell; CLP: common lymphoid progenitor; GMP: granulocyte-macrophage progenitor; CMP: common myeloid progenitor; MEP: megakaryocyte-erythroid progenitor; Mk-P: megakaryocyte progenitor; Sl-Mkp: stem-like megakaryocyte progenitors; TPO: thrombopoietin; MK: megakaryocyte; IL: interleukin.

**Figure 2 ijms-24-01010-f002:**
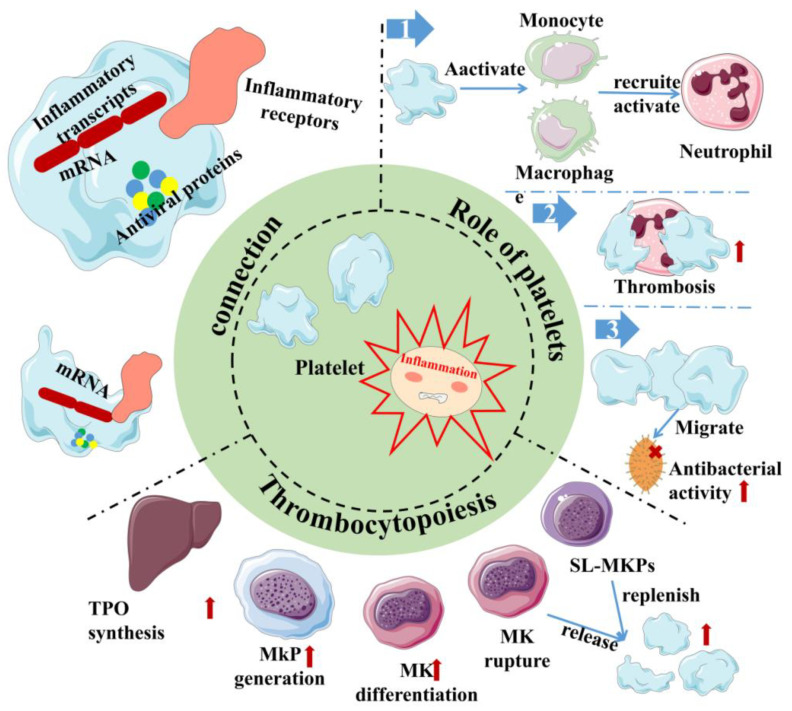
Inflammation and platelet interaction. Platelets are at the junction between inflammation and host defense. Platelets carry not only inflammatory transcripts but also antiviral proteins. Platelets activate/recruit a variety of cells for host defense in inflammation, meanwhile, and inflammation can promote platelet production through multiple channels. Red arrows indicate the enhanced physiological effects. Red “×” indicate the role of sterilization. Sl-Mkp: Stem-like megakaryocyte progenitors; TPO: Thrombopoietin; MK: megakaryocyte; MkP: Megakaryocyte progenitor.

**Figure 3 ijms-24-01010-f003:**
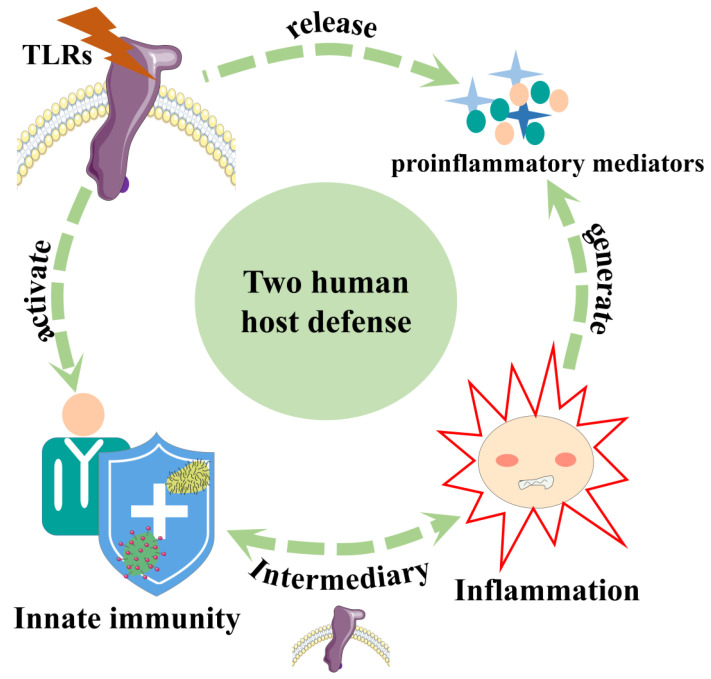
Inflammation and TLRs. Innate immunity and inflammation are two major defense mechanisms in the body. Activation of TLRs releases inflammatory factors, and TLRs are an important mediators between the inflammatory response and innate immunity.

**Figure 4 ijms-24-01010-f004:**
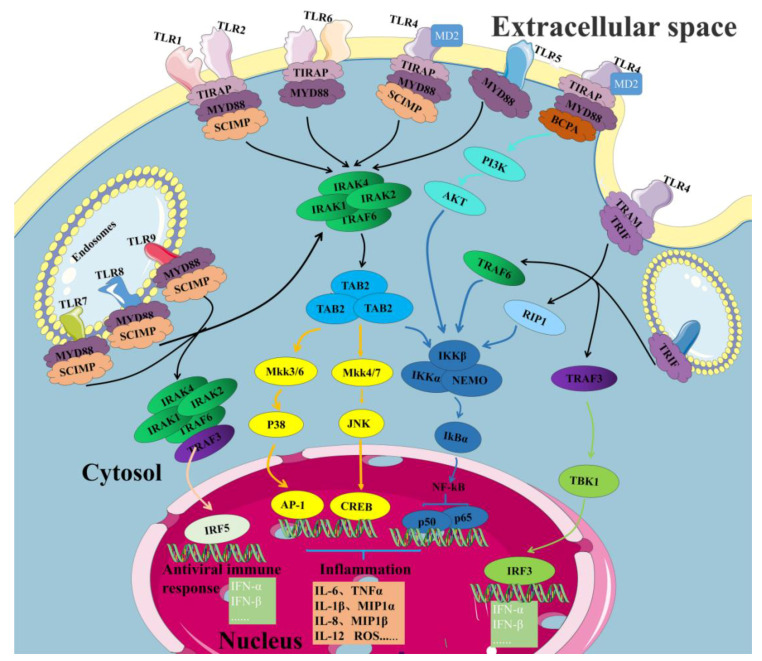
TLR signaling pathway. TLRs can be expressed on the surface of cells or inside cells. However, TLRs can activate corresponding signaling pathways through different ligands, and ultimately, they all focus on some common downstream signaling pathways to induce the production of pro-inflammatory cytokines and type I interferon. MD2: Myeloid differentiation protein 2; IRAK: Interleukin-1 receptor-associated kinase; TRAF: TNF receptor-associated factor; PI3K: RIP1: Receptor-interacting protein 1; TAB: TGF-beta-activated kinase 1 and MAP3K7-binding protein; IKKα: Inhibitor of nuclear factor kappa-B kinase subunit alpha; IKKβ: Inhibitor of nuclear factor kappa-B kinase subunit beta; IKBα: I-kappa-B-alpha; NEMO: NF-kappa-B essential modifier; TBK1: Serine/threonine-protein kinase TBK1; for other abbreviations, see Table 1.

**Figure 5 ijms-24-01010-f005:**
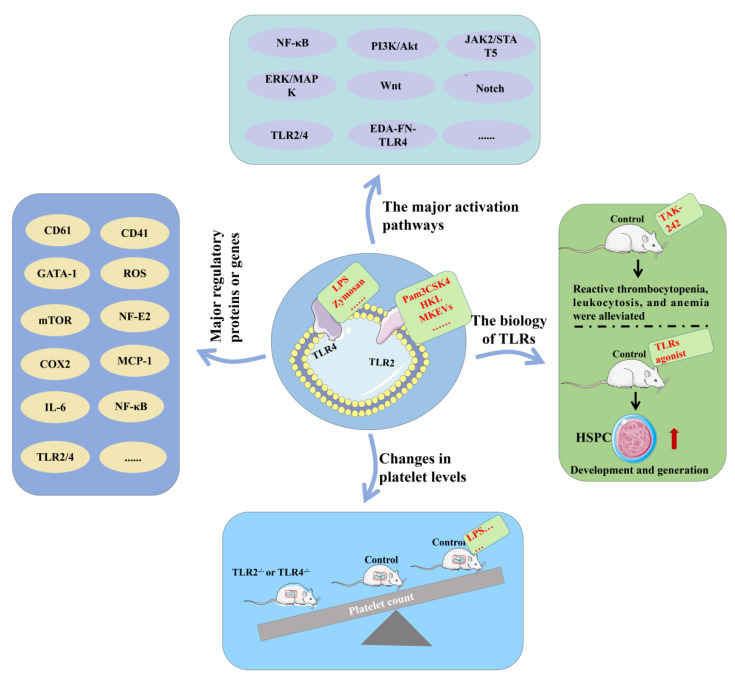
TLR2 and TLR4 promote platelet formation. TLR activation can activate multiple signaling pathways and increase the expression of related factors, thereby increasing the number of platelets. In vivo TLR inhibitor studies further confirmed that TLR activation can affect platelet production.

**Figure 6 ijms-24-01010-f006:**
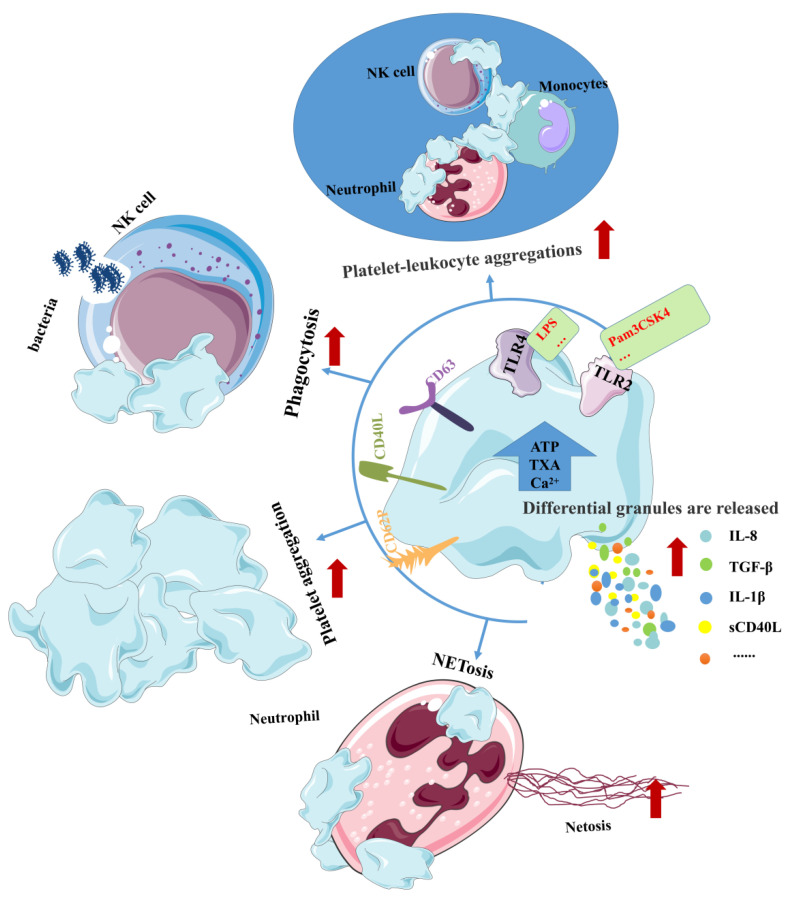
TLR2 and TLR4 increase platelet function. The activation of TLR on platelets can release a large number of inflammatory factors, activate and aggregate platelets, and recruit other cells to aggregate or carry out anti-inflammatory, antibacterial and other effects. Red arrows indicate the enhanced physiological effects. NK cell: natural killer Cell; LPS: Lipopolysaccharides; ATP: adenosine triphosphate; TXA: Tranexamic Acid; TGF-β: Transforming Growth Factor Beta; sCD40 L: soluble CD40-ligand; NETosis: neutrophil extracellular trap (NET) formation.

**Figure 7 ijms-24-01010-f007:**
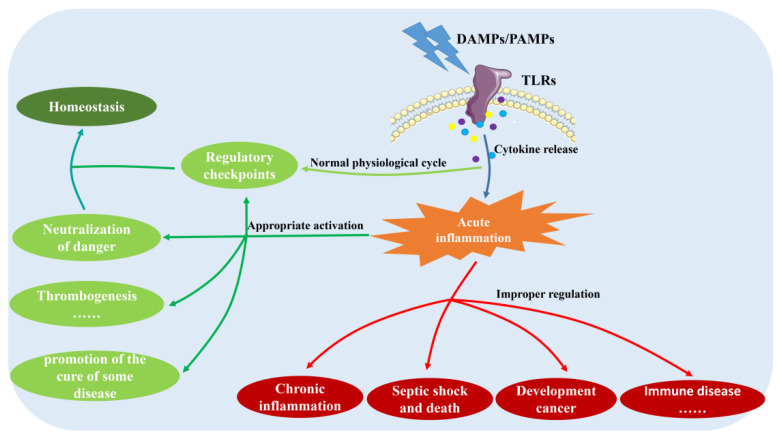
The importance of balancing the TLR2 and TLR4 pathways. TLR activation can restore homeostasis via regulatory checkpoints. We can also use TLR activation to produce inflammatory factors to achieve the therapeutic effect of diseases, but it is worth noting that once over-activated, it will also lead to a variety of diseases. Therefore, it is critical to grasp the extent of TLR activation. DAMPs: Danger-associated molecular patterns; PAMPs: Pathogen-associated molecular patterns; TLRs: Toll-like receptors.

**Table 1 ijms-24-01010-t001:** Classification of TLR-related adapters.

Typical Aptamer	Regulatory Aptamer
myeloid differentiation primary-response protein 88 (MyD88)	Sterile α and armadillo motif-containing protein (SARM)
TIR-domain-containing adaptor protein (TIRAP), also known as MyD88 adapter-like protein (MAL)	B-cell adaptor for phosphoinositide 3-kinase (BCAP), also known as Phosphoinositide 3-kinase adapter protein 1 (PIK3AP1)
TIR-domain-containing adaptor protein inducing interferon-β (TRIF), also known as TIR domain-containing adapter molecule 1 (TICAM1)	SLP adapter and Csk-interacting membrane protein(SCIMP)
TRIF-related adaptor molecule (TRAM) also known as TIR domain-containing adapter molecule 2 (TICAM2)	

**Table 2 ijms-24-01010-t002:** Some experimental types of TLR agonists.

Tagonist	Target TLR	Biological Activity	References
LPS/lipid A	TLR4	It is a compound of lipids and polysaccharides. This product is lipopolypaccharide purified from E. coli O111:B4, which can specifically activate TLR4 but not TLR2.	
Paclitaxel	TLR4	Paclitaxel is one of the confirmed direct mouse TLR4/MD-2 agonists and is an antagonist of the human TLR4 receptor complex	[161]
Ni^2+^ ions	TLR4	Specific activation of human TLR4/MD-2 by Ni^2+^, but not of the mouse receptor	[161]
CU-T12-9	TLR2	CU-T12-9 is a specific TLR1/2 agonist with EC50 of 52.9 nM in HEK-Blue hTLR2 SEAP assay. CU-T12-9 activates both the innate and the adaptive immune systems. CU-T12-9 selectively activates the TLR1/2 heterodimer, not TLR2/6. CU-T12-9 signals through NF-κB and invokes an elevation of the downstream effectors TNF-α, IL-10, and iNOS	[162]
SMU-Z1	TLR2	SMU-Z1 exhibited specific activation of TLR1/TLR2 signaling and showed antitumor immunity against leukemia in a murine leukemia model.	[163]
FSL-1	TLR2	FSL-1, a bacterial-derived toll-like receptor 2/6 (TLR2/6) agonist, enhances resistance to experimental HSV-2 infection.	[164,165]
FSL-1 TFA	TLR2	FSL-1 TFA, a bacterial-derived toll-like receptor 2/6 (TLR2/6) agonist, enhances resistance to experimental HSV-2 infection. FSL-1 TFA induces MMP-9 production through TLR2 and NF-κB/AP-1 signaling pathways in monocytic THP-1 cells	[164,166]
GSK1795091 (CRX-601)	TLR4	GSK1795091 (CRX-601), an immunologic stimulator, is a synthetic TLR4 agonist. Antitumor activity. GSK1795091 can be used as a vaccine adjuvant to enhance both mucosal and systemic immunity to influenza virus vaccines	[167,168]
RS 09 TFA	TLR4	RS 09 TFA is a TLR4 agonist. RS 09 TFA promotes NF-κB nuclear translocation and induces inflammatory cytokine secretion in RAW264.7 macrophages in vitro. RS 09 TFA acts as an adjuvant in vivo; RS 09 TFA enhances X-15 specific antibody serum concentrations, when administered with X-15-KLH in mice	[169,170]
HMGB1	TLR5	HMGB1 can bind to TLR5 to initiate its downstream NF-κB signaling pathway activation and induce proinflammatory cytokine	[170]
Imiquimod	TLR7	Imiquimod is the first FDA-approved agonist targeting TLR7 for the treatment of external genital warts.	[171]

**Table 3 ijms-24-01010-t003:** Some experimental types of TLR antagonists.

Antagonist	Target TLR	Mechanism	References
TAK-242 (Resatorvid)	TLR4	TAK-242 binds selectively to TLR4 (Cys747) and subsequently disrupts the interaction of TLR4 with adaptor molecules, thereby inhibiting TLR4 signal transduction and its downstream signaling events	[172]
E5564 (Eritoran)	TLR4	E5564 (a novel Toll-like receptor 4-directed endotoxin antagonist) can block TLR4 activation through prevention of LPS binding to the TLR4-MD2 complex and lack agonistic activity in human and animal model systems, making it a potentially effective therapeutic agent for treatment of disease states caused by endotoxin.	[173]
OPN-305	TLR2	OPN-305 is the first humanized IgG4 monoclonal antibody against TLR2 in development and is intended for the prevention of reperfusion injury following renal transplantation and other indications.	[174]
CRX-526	TLR4	CRX-526, which has antagonistic activity for TLR4 and can block TLR4 activation through prevention of LPS binding to the TLR4-MD2 complex. CRX-526 can prevent the expression of proinflammatory genes stimulated by LPS in vitro	[175]
IAXO-102	TLR4	IAXO-102 is a TLR4 antagonist which negatively regulates TLR4 signaling. IAXO-102 inhibits MAPK and p65 NF-κB phosphorylation and expression of TLR4 dependent proinflammatory protein. IAXO-102 also prevents experimental abdominal aortic aneurysm development	[176]
lovastatin	TLR4	lovastatin may be a potential drug to be repurposed for treating chronic pain	[177]
CU-CPT22	TLR2	CU-CPT22 is a potent protein complex of toll-like receptor 1 and 2 (TLR1/2) inhibitor, and competes with the synthetic triacylated lipoprotein (Pam3CSK4) binding to TLR1/2.	[178]
CAY10614	TLR4	CAY10614 is a potent TLR4 antagonist. CAY10614 inhibits the lipid A-induced activation of TLR4.	[179,180]
(+)-norbinaltorphimine	TLR4	the TLR4 antagonistic activity of (+)-norbinaltorphimine increased to 4.7 ± 1.8 μM in the NO assay and significantly enhanced and prolonged morphine analgesia in vivo.	[181]
NI-0101	TLR4	Primarily blocking THE dimerization of TLR4 and blocking the production of pro-inflammatory cytokines in synovial stimulated monocytes in RA patients, it has been tested in clinical trials in RA patients but unfortunately has not shown any benefit	[182]
C_16_H_15_NO_4_ (C29)	TLR2	C29, and its derivative, ortho-vanillin (o-vanillin), inhibited TLR2/1 and TLR2/6 signaling induced by synthetic and bacterial TLR2 agonists in human HEK-TLR2 and THP-1 cells, but only TLR2/1 signaling in murine macrophages.	[183]
Hydroxychloroquine	TLR7/9	Hydroxychloroquine is an autophagy inhibitor, which may target TLR7 and TLR9.	[184]
SSL3	TLR2	SSL3 inhibits binding of bacterial lipopeptides, and, second, if a lipopeptide has already been engaged by TLR2, SSL3 prevents the formation of TLR2–TLR1 and TLR2–TLR6 heterodimers.	[185]
CXC195	TLR4	CXC195 Exerts anti-proliferative effects through TLR4-mediated suppression of inflammatory cytokines.	[186]
Atractylenolide	TLR4	Atractylenolide can make TLR4 and MyD88/NF-κB in ovarian cancer cells downregulatie	[187]
Triptolide	TLR4	Triptolide can inducie suppressive effect on the TLR4/NF-κB axis.	[188]
Paeonol	TLR4	Paeonol can abolish the propagation of the TLR4/MAPK/NF-κB signaling axis.	[189]
MMG-11	TLR2	MMG-11 is a potent and selective human TLR2 antagonist with low cytotoxicity. MMG-11 inhibits both TLR2/1 and TLR2/6 signaling.	[190]

## Data Availability

Not applicable.

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
