# Peer review of "Toll-like Receptors and Thrombopoiesis"

_ijms, 2023, doi:10.3390/ijms24021010_

Round 1

Reviewer 1 Report

Comment to authors:

The presented review manuscript, entitled: Toll-like receptors and thrombopoiesis, reviews the importance of TLRs in the context of thrombopoiesis and inflammatory insults. While the reviewer does appreciate the effort, the following major concerns are raised, and would be appreciated if addressed.

Major Comments:

1.     First paragraph, lines 35-48: The authors should try to have this part of the introduction written in a more focused fashion. While the general message of this paragraph describes megakaryo- and thrombopoiesis, more references would be appreciated, and the lack of scientific language and rigor, as exemplified in line 47 (“…and so on.”) should be addressed and fixed.

2.     First paragraph, lines 48-55: This part of the introduction misses references at all, which should be fixed, and the appropriate and relevant references should be introduced.

3.     All parts: Please support your conclusions and claims using the appropriate references.

4.     All parts: The reviewer would appreciate an in-depth analysis of the wording, careful choosing of such, and making sure that repetitive and/or misleading conclusions are consolidated.

5.     All parts: The reviewer would appreciate if non-scientific wording could be deleted from figures, and figures could be appropriately re-designed.

6. All parts: The manuscript would overall benefit from some stream-lining in regards of thought process, language, content, and conclusions.

Author Response

Dec. 27, 2022

Dear Expert Reviewer,

Thank you very much for the prompt review process and excellent comments. We greatly appreciate the time and efforts which you have spent on it. We are submitting the revised manuscript entitled “Toll-like receptors and thrombocytopoiesis” (ID: ijms-2121528) to International Journal of Molecular Sciences.

We have carefully considered your comments and suggestions, and addressed each of the concerns in response to the comments (see point by point response). We have revised the manuscripts based on your comments and carefully checked throughout the manuscript and corrected the language errors. Our point-by-point responses to the comments (in blue) are shown below (in red).

  1. First paragraph, lines 35-48: The authors should try to have this part of the introduction written in a more focused fashion. While the general message of this paragraph describes megakaryo- and thrombopoiesis, more references would be appreciated, and the lack of scientific language and rigor, as exemplified in line 47 (“…and so on.”) should be addressed and fixed.

Response: Thanks a lot for the excellent comments and suggestions. According to your suggestions, we have rewrote the contents of megakaryo- and thrombopoiesis and added some references in revised manuscript (page 1, line 35-47).

  1. First paragraph, lines 48-55: This part of the introduction misses references at all, which should be fixed, and the appropriate and relevant references should be introduced.

Response: Thank you very much for your thoughtful advice. We have added the appropriate and relevant references ( page 2, line 53, 56).

  1. All parts: Please support your conclusions and claims using the appropriate references.

Response: Thank you for your careful reading. In the revised manuscript, we have inserted the appropriate references to support our conclusions and claims (page 3, line 96, 98, 123, 129, 132; page 4, line 134, 159; page 5, line 186; page 6, line 219, 223; page 7, line 254; page 10, line 388).

  1. All parts: The reviewer would appreciate an in-depth analysis of the wording, careful choosing of such, and making sure that repetitive and/or misleading conclusions are consolidated.

Response: Thank you very much for your carefully reading and kindheartedly reminder. We carefully checked the whole manuscript and revised some of words in manuscript. Meanwhile, our English editting have been polished by American Journal Experts. (page 2, line 70; page 3, line 81, 107, 125; page 5, line 180, 190, 195, 198; page 6, line 235, 238, 239; page 7, line 247, 252, 257; page 8, line 294; page 9, line 305, 348; page 10, line 363, 369, 372, 374, 384-390, 405, 408; page 11, line 428; page 12, line 441; page 14, line 484, 504; page 15, line 539, 544, 548, 552; page 16, line 574; page 17, line 586, 588-589, 592-593, 609, 612, 617, 627; page 18, line 634,636-637, 639-642, 644).

  1. All parts: The reviewer would appreciate if non-scientific wording could be deleted from figures, and figures could be appropriately re-designed.

Response: Thank you very much for your excellent comments and suggestions. According to your suggestions, we have revised non-scientific wording in some figures. (Figure 3.5.6.7)

  1. All parts: The manuscript would overall benefit from some stream-lining in regards of thought process, language, content, and conclusions.

Response: Thanks for your careful reading. We have revised the manuscripts based on your comments and carefully checked throughout the manuscript and corrected the language errors.

Thank you for all the valuable and helpful comments and suggestions.

Best regards,

Jianming Wu, Ph.D & Professor

Reviewer 2 Report

The manuscript: Toll-like receptors and thrombopoiesis by Xiaoqin Tang et al. is a very interesting review article discussing the relationship between TLRs and thrombopoiesis. The review is very interesting and valuable. The article contains many interesting figures, presenting in a legible and clear way the processes in which TLRs participate. The structure of the manuscript is also well thought out and contains a logical sequence characteristic of a scientific article. In my opinion, the article can be published, but before that happens, I have a few minor comments, which I present below:

1. In order to improve the readability and pleasure of reading the article, I would suggest adding abbreviations to the descriptions of the figures, e.g. Figure 1. - GMC, CLP, MEP, HSC, etc. Similarly in the other figures

2. In the whole text, there are often no spaces before citation numbers, e.g. lines: 44, 63,67, 70, 75 etc.

3. In order to improve the readability of tables 2, 3, I would suggest adding an additional column at the bottom in which reference numbers would be entered,

4. As in the case of figures, I would suggest expanding the abbreviations in the description of table 1.

5. The article contains 174 citations, I would like to know what inclusion or exclusion criteria the authors used when searching for articles? What keywords were used. I would like to add this information to the article. The authors may suggest the PRISMA statement methodology. Authors can provide a diagram that would show the procedure for searching for articles (https://www.prisma-statement.org/)

Author Response

Dec. 27, 2022

Dear Expert Reviewer,

Thank you very much for the prompt review process and excellent comments. We greatly appreciate the time and efforts which you have spent on it. We are submitting the revised manuscript entitled “Toll-like receptors and thrombocytopoiesis” (ID: ijms-2121528) to International Journal of Molecular Sciences.

We have carefully considered your comments and suggestions, and addressed each of the concerns in response to the comments (see point by point response). We have revised the manuscripts based on your comments and carefully checked throughout the manuscript and corrected the language errors. Our point-by-point responses to the comments (in blue) are shown below (in red).

  1. In order to improve the readability and pleasure of reading the article, I would suggest adding abbreviations to the descriptions of the figures, e.g. Figure 1. - GMC, CLP, MEP, HSC, etc. Similarly in the other figures

Response: Thank you very much for your thoughtful advice. We have added some abbreviations to the descriptions of the figures (page 1, lines 64-68; page 3, lines 153-155; page 9, lines 307-313; page 13, lines 461-464; page 15, lines 533-535).

  1. In the whole text, there are often no spaces before citation numbers, e.g. lines: 44, 63,67, 70, 75 etc.

Response: Thank you very much for your carefully reading and kindly remindering. We carefully checked the whole manuscript and added spaces before citation numbers.

  1. In order to improve the readability of tables 2, 3, I would suggest adding an additional column at the bottom in which reference numbers would be entered.

Response: Thanks a lot for the constructive and careful suggestion. We have added an additional column at the bottom and entered the reference numbers (page 16, lines 576; page 16-17, lines 577).

  1. As in the case of figures, I would suggest expanding the abbreviations in the description of table 1.

Response: Thank you for your rigorous thinking. We have refined the abbreviations in the description of Table 1 (page 6-7, lines 285).

  1. The article contains 174 citations, I would like to know what inclusion or exclusion criteria the authors used when searching for articles? What keywords were used. I would like to add this information to the article. The authors may suggest the PRISMA statement methodology. Authors can provide a diagram that would show the procedure for searching for articles (https://www.prisma-statement.org/)

Response: Thanks a lot for the excellent comments and suggestions. According to your suggestions, we have added PRISMA diagram in the supplementary materials.

Thank you for all the valuable and helpful comments and suggestions.

Best regards,

Jianming Wu, Ph.D & Professor

Round 2

Reviewer 1 Report

Dear Authors,

the reviewer much appreciates your efforts in improving the current manuscript. The addition of significant references strengthened the conclusions, and will help the readership to understand the context in light of previous findings in the field. The reviewer also much appreciates the improved language, the revised figures and the editing efforts.